# Genomic Features and Clinical Management of Patients with Hereditary Pancreatic Cancer Syndromes and Familial Pancreatic Cancer

**DOI:** 10.3390/ijms20030561

**Published:** 2019-01-29

**Authors:** Akihiro Ohmoto, Shinichi Yachida, Chigusa Morizane

**Affiliations:** 1Laboratory of Clinical Genomics, National Cancer Center Research Institute, Tokyo 1040045, Japan; syachida@cgi.med.osaka-u.ac.jp; 2Department of Cancer Genome Informatics, Graduate School of Medicine/Faculty of Medicine, Osaka University, Osaka 5650871, Japan; 3Department of Hepatobiliary and Pancreatic Oncology, National Cancer Center Hospital, Tokyo 1040045, Japan; cmorizan@ncc.go.jp

**Keywords:** hereditary cancer syndrome, Lynch syndrome, familial pancreatic cancer, next-generation sequencing, germline mutation, surveillance, PARP inhibitor, immune checkpoint inhibitor

## Abstract

Pancreatic cancer (PC) is one of the most devastating malignancies; it has a 5-year survival rate of only 9%, and novel treatment strategies are urgently needed. While most PC cases occur sporadically, PC associated with hereditary syndromes or familial PC (FPC; defined as an individual having two or more first-degree relatives diagnosed with PC) accounts for about 10% of cases. Hereditary cancer syndromes associated with increased risk for PC include Peutz-Jeghers syndrome, hereditary pancreatitis, familial atypical multiple mole melanoma, familial adenomatous polyposis, Lynch syndrome and hereditary breast and ovarian cancer syndrome. Next-generation sequencing of FPC patients has uncovered new susceptibility genes such as *PALB2* and *ATM*, which participate in homologous recombination repair, and further investigations are in progress. Previous studies have demonstrated that some sporadic cases that do not fulfil FPC criteria also harbor similar mutations, and so genomic testing based on family history might overlook some susceptibility gene carriers. There are no established screening procedures for high-risk unaffected cases, and it is not clear whether surveillance programs would have clinical benefits. In terms of treatment, poly (ADP-ribose) polymerase inhibitors for *BRCA*-mutated cases or immune checkpoint inhibitors for mismatch repair deficient cases are promising, and clinical trials of these agents are underway.

## 1. Introduction

Pancreatic cancer (PC) is one of the most devasting malignancies worldwide. In spite of advances in early detection, surgical techniques and new systemic treatment, the 5-year survival rate in PC patients is still only about 9% [1]. According to the Global Cancer Statistics 2018, PC is ranked as the seventh leading cause of cancer-related mortality, accounting for 4.5% of all malignancy cases [2]. Previous studies have implicated lifestyle-related factors such as smoking, heavy alcohol consumption, obesity and diabetes mellitus in PC onset, and most PC cases develop sporadically [3]. Thus, identifying patients with predisposing genetic factors seems an attractive strategy for improving clinical outcomes. It is estimated that 3% of PC cases derive from hereditary cancer syndromes, and another 7% of the cases are classified as familial PC (FPC), which is defined as an individual who has two or more first-degree relatives (FDRs) with PC [4,5].

Clinical management for PC probands necessarily raises the problem of surveillance, including counseling, for unaffected relatives. The main purpose of surveillance for high-risk relatives is the detection of precursor lesions or early PC, which is the only point at which a curative approach may be feasible at present. However, standard screening procedures have not been settled, and it is not yet clear whether screening programs offer clinical benefits. As regards treatment, molecular-targeted agents or immune therapies are rapidly being introduced into clinical practice even though their usefulness for PC is not established. However, favorable data has emerged in clinical trials of poly (ADP-ribose) polymerase (PARP) inhibitors for *BRCA*-mutated cases or anti-PD-1 antibodies for mismatch repair (MMR)-deficient cases. Here, we present an overview of the latest studies on PC associated with hereditary cancer syndromes and FPC, focusing on genomic data, and we describe the current status and future prospects of surveillance programs and therapeutic interventions.

## 2. Hereditary Pancreatic Cancer Syndromes

Several hereditary syndromes are associated with the onset of PC. The 2015 clinical guideline for hereditary gastrointestinal cancer syndromes by the American College of Gastroenterology (ACG) covers Peutz-Jeghers syndrome (PJS), hereditary pancreatitis (HP), familial atypical multiple mole melanoma (FAMMM), familial adenomatous polyposis (FAP), Lynch syndrome (LS) and hereditary breast and ovarian cancer syndrome (HBOC), together with Li-Fraumeni syndrome harboring *TP53* mutation or ataxia-telangiectasia harboring *ATM* mutation [6]. Representative hereditary syndromes with specific germline mutations are summarized in Table 1.

### 2.1. Peutz-Jeghers Syndrome (PJS)

PJS is an autosomal dominant inherited syndrome. Perioral/buccal pigmentation and gastrointestinal multiple hamartomatous polyps are characteristic of PJS, and these symptoms are seen in more than 95% and 88–100% of cases, respectively [6]. Pathologically, intraductal papillary mucinous neoplasm (IPMN) as a precursor lesion of pancreatic ductal adenocarcinoma (PDAC) is observed in some patients with PJS [7]. Inactivating mutations of the *STK11*/*LKB1* gene regulating cell growth, proliferation and DNA damage response are causative for PJS [8]. This syndrome increases the incidence of PC, as well as gastrointestinal, lung, breast, uterus and ovarian cancers [9]. A meta-analysis covering 210 PJS cases showed a 15.2-fold increased risk of any cancer, and the risk was especially high for PC (132-fold), as well as esophageal cancer (57-fold), stomach cancer (213-fold) and small intestinal cancer (520-fold) [9]. Hearle et al., reported that 297 of 419 (70.9%) cases with PJS harbored *STK11*/*LKB1* mutation, and cumulative risks of any cancer and PC were 85% and 11%, respectively, at 70 years of age [10]. Here, the cumulative risk of any cancer was not significantly different between the cases with *STK11*/*LKB1* mutation and those without the mutation (*p* = 0.43). In an analysis of a series of 240 PJS patients harboring *STK11* mutation, cumulative PC risk was 8% at 60 years of age [11]. These data highlight PJS as one of the highest risk factors for PC onset.

### 2.2. Hereditary Pancreatitis (HP)

Patients with HP suffer from recurrent acute pancreatitis, leading eventually to chronic pancreatitis. Because sporadic chronic pancreatitis usually occurs in elderly people, early onset (age < 25 years) is a clue for suspecting HP [12]. Activating mutations in *PRSS1* encoding the cationic trypsinogen related to trypsin activation, and inactivating mutations in *SPINK1* inhibiting trypsin are causative for this syndrome [13]. It is suggested that repeated mechanical damage to acinar cells due to continuous trypsin activation induces PC onset. Pancreatic cancer deriving from HP usually exhibits the pathology of typical PDAC [7]. Lowenfels et al. reported that HP increased PC risk by 53-fold and the cumulative PC risk reached 40% at 70 years of age [14]. An analysis of a series of 200 HP patients showed 87-fold increased PC risk, and cumulative PC risk was 53.5% at 75 years of age [15]. According to a recent study of 217 *PRSS1* mutation carriers, R122H and N29I variants were detected in 83.9% and 11.5% of the cases, respectively, and cumulative PC risk was 7.2% at 70 years of age [16]. The cumulative risk estimated in this study is much lower than those found by Lowenfels et al. and Rebours et al. [14,15], though this might be explained by different patient backgrounds, referral bias or lifestyle changes including smoking. In genomic analysis of a series of 41 Polish children with HP, Oracz et al. detected *PRSS1* mutation in 80.5% of the cases (34% R122H variant; 27% R122C; 12% N29I; 7% E79K) [17]. Rebours et al. also conducted a genomic analysis of a cohort of 200 French HP cases, and detected *PRSS1* mutation in 68% (53% R122H variant; 8% N29I) and *SPINK1* mutation in 13% of the cases [18]. A Japanese survey of 271 patients in 100 HP families showed a *PRSS1* mutation prevalence of 41.1% and a *SPINK1* mutation prevalence of 35.6% [19]. The decreased ratio of *PRSS1*-mutated cases compared with the western cohort might reflect different genomic backgrounds among ethnic groups [17,18,19]. As with PJS, HP is categorized as one of the highest risk factors for PC.

### 2.3. Familial Atypical Multiple Mole Melanoma (FAMMM)

Patients with FAMMM have multiple (usually > 50) atypical nevi progressing to melanoma [20]. FAMMM leads to increased incidences of PC as well as breast, lung and endometrium cancers [12]. Inactivating mutations in *CDKN2A* (*p16*) inducing G_1_/G_2_ cell cycle arrest are causative for this syndrome [8]. PC deriving from FAMMM is pathologically ordinary [7]. Vasen et al. showed that cumulative PC risk in FAMMM families harboring *CDKN2A* mutation was 17% at 75 years of age [21]. Goldstein et al. and De Snoo et al. reported that relative PC risk in families with *CDKN2A* mutation was 13.1–22 and 46.6, respectively [22,23]. While self-monitoring of nevi is useful in detecting melanoma, surveillance procedures for detecting pancreatic lesions early have not been fully established [24]. Vasen et al. monitored 77 germline *CDKN2A* mutation carriers with magnetic resonance imaging (MRI) or magnetic resonance cholangiopancreatography (MRCP), and detected 7 (9.1%) resectable PC cases, concluding that further studies are warranted [25].

### 2.4. Familial Adenomatous Polyposis (FAP)

FAP is characterized by hundreds of synchronous colorectal adenomas, and these adenomas inevitably progress into malignancies at an average age of 35–40 years [26]. *APC* regulating cell migration and adhesion and *MUTYH* participating in base excision repair are causative genes for FAP [8]. Giardiello et al. reported that FAP increased PC risk by 4.5-fold and cumulative PC risk was 1.7% at 80 years of age [27]. A pathological review of four PC cases deriving from FAP found that all of them exhibited unusual histology (poorly differentiated neuroendocrine carcinoma, acinar cell carcinoma and pancreatoblastoma) [28]. Among them, gene mutations in the *APC*/β-catenin pathway were detected in pancreatoblastoma, indicating a molecular interaction between FAP and pancreatoblastoma [29].

### 2.5. Lynch Syndrome (LS)

Patients with LS suffer from colorectal cancer at early ages (typically in the mid 40 years, which is about 20 years younger than sporadic cases). LS is also termed hereditary nonpolyposis colorectal cancer, and is genetically characterized by the presence of inactivating mutations in MMR genes (*MLH1*, *MSH2*, *MSH6* and *PMS2*) and *EPCAM* [6]. *EPCAM* was newly discovered as a causative gene for LS, and 3′ end deletion of this gene causes epigenetic silencing of the *MSH2* gene in EPCAM-expressing tissue [30]. The revised Bethesda Guidelines (RBG) for identifying MMR gene mutation carriers categorize PC as one of the LS-associated neoplasms, along with other types of malignancies such as endometrial, small intestinal or ureter/renal pelvic cancers [31]. Pathologically, medullary carcinoma, a rare subtype, is uniquely observed in LS-derived PC [7]. Kastrinos et al. analyzed 6,342 PC probands and relatives from 147 families harboring MMR gene mutations, and found that this syndrome increased PC risk by 8.6-fold and cumulative PC risk was 3.7% at 70 years of age [32]. A recent prospective observational study of 3119 MMR-mutation carriers found that relative cumulative PC risk was 7.8 (95% CI: 3.3–12.3) in *MLH1* mutation carriers at 75 years of age, although *MSH2*, *MSH6* or *PMS2* carriers did not show increased risk [33].

Considering the worldwide clinical investigations of immune checkpoint inhibitors, identifying MMR-mutated cases has become practically more important. Our group selected 20 (6.6%) patients with a personal or family history consistent with RBG from a cohort of 304 Japanese PC patients, and analyzed germline variants of 21 hereditary cancer susceptibility genes [34]. We detected *PMS2* mutation in only one case (0.3% of all 304 cases). Hu et al. exhaustively conducted immunohistochemistry (IHC), microsatellite instability (MSI) testing and germline DNA sequencing in a series of 833 PC patients, and identified 7 (0.8%) MMR mutation carriers, including 5 cases meeting RBG criteria [35]. Because of the lower lifetime PC risk (1–6%) compared with colorectal cancer (10–82%) or endometrial cancer (15–60%), clinical evidence that universal tumor screening would be effective in patients with LS, as recommended for colorectal or endometrial cancers, is scarce for PC [36]. In order to find favorable candidates for immunotherapy, Hu et al. proposed a practical algorithm for efficiently extracting MMR-deficient PC cases, in which IHC or MSI testing is initially assigned to resected cases, and next-generation sequencing to metastatic/locally advanced cases [35].

### 2.6. Hereditary Breast and Ovarian Cancer Syndrome (HBOC)

HBOC is a syndrome characterized by breast and ovarian cancers occurring before the age of 50, and accounts for 5–10% of breast cancers and 10–15% of ovarian cancers [37,38]. HBOC is genetically caused by inactivating mutations in *BRCA1*/*BRCA2*, which are involved in the homologous recombination repair (HRR) pathway. While *BRCA2* mutation is widely accepted as a PC risk factor, the data for *BRCA1* mutation are conflicting [39]. Iqbal et al. prospectively identified 8 PC cases from 3942 *BRCA1* and 1147 *BRCA2* mutation carriers, and reported that the relative PC risk compared with the general population was 2.6 (95% CI: 1.0–5.3) in *BRCA1* carriers and 2.1 (95% CI: 0.4–7.0) in *BRCA2* carriers [40]. According to a study of 613 *BRCA1* and 459 *BRCA2* mutation carriers, relative PC risk for *BRCA2* mutation carriers was 21.7 (95% CI: 13.1–34.0), whereas *BRCA1* mutation did not significantly increase the risk [41]. As Klein noted, the above studies did not establish whether a past history of breast or ovarian cancer is essential to the diagnosis of HBOC. Therefore, it is not clear whether PC risk is similar between *BRCA*1/2 carriers fulfilling the criteria of HBOC and those not meeting the criteria [3].

## 3. Familial Pancreatic Cancer (FPC) and Susceptibility Genes

As mentioned above, FPC refers to individuals having two or more FDRs with PC, and patients associated with hereditary syndromes are excluded from this category [5]. From the viewpoint of clinicopathological features, Singhi et al. compared 519 FPC and 651 sporadic PC, and reported that there was no significant difference in histological subtype, patient age, tumor size, tumor location, peripheral invasion, angiolymphatic invasion, lymph node metastasis or pathological stage [42]. According to a prospective cohort study of 5,179 individuals from 838 families, the relative PC risk was 4.5 (95% CI: 0.5–16.3) in 1,253 cases with one affected FDR, 6.4 (95% CI: 1.8–16.4) in 634 cases with two affected FDRs, and 32.0 (95% CI: 10.4–74.7) in 106 cases with three or more affected FDRs [43]. A meta-analysis including seven case-control and two cohort studies demonstrated that an individual with an affected relative, irrespective of the degree of relationship, had an increased PC risk of 1.8 (95% CI: 1.5–2.1) [44]. A pooled analysis from the Pancreatic Cancer Cohort Consortium also showed that an individual with a family history of PC in an FDR had an increased PC risk of 1.8 (95% CI: 1.2–2.6) [45]. These reports confirm that the existence of an affected FDR is a significant PC risk, especially where there is a strong family history.

Next-generation sequencing has contributed to the discovery of novel FPC susceptibility genes such as *PALB2* or *ATM*. Initially, Murphy et al. identified germline *BRCA2* mutation in 5 of 29 (17.2%) FPC patients, including 3 cases harboring 6174delT frameshift variant [46]. Hahn et al. also identified a germline *BRCA2* frameshift variant in 3 of a cohort of 26 (11.5%) FPC patients [47]. It is known that *BRCA2* mutation prevalence differs among ethnic groups, and is especially high in Ashkenazi Jews. Germline mutation analysis of 5,318 Jewish subjects detected *BRCA2* 6174delT variant in 1.2%, as well as *BRCA1* 185delAG or 5382insC variant in 1.2% [48]. In contrast to *BRCA2*, the involvement of *BRCA1* in FPC is controversial. Indeed, germline *BRCA1* mutation analysis of 66 PC patients with 2 or more affected relatives detected no mutated cases [49]. As a second susceptibility gene, Jones et al. discovered a *PALB2* frameshift variant (c.172_175 delTTGT) in one FPC patient by examining whole-exome sequencing data, and identified *PALB2* truncation mutation leading to a stop codon in 3 of 96 (3.1%) FPC cases, in comparison with no mutation in the control cohort of 1,084 [50]. Interestingly, the locations of mutations found in this study were different from those previously reported in familial breast cancer or Fanconi anemia [50,51]. Slater et al. also sequenced the 13 exons of the *PALB2* gene for 81 FPC families, and identified truncating mutation leading to a stop codon in 3 (3.7%) cases [52]. Furthermore, Roberts et al. discovered heterozygous *ATM* nonsense variants (c. 8266A>T; c.170G>A) in 2 FPC kindreds from whole-genome or whole-exome sequencing data, and validated deleterious *ATM* mutations in 4 of 166 (2.4%) FPC probands in comparison with no mutation in a control cohort [53]. When the subjects were restricted to 87 probands with more than three affected members, *ATM* mutation prevalence was 4.6%. BRCA2, PALB2 and ATM are all involved in the HRR pathway. In brief, ATM is recruited in response to DNA double-strand breaks (DSBs) induced by DNA damage and activates CHK2. PALB2 promotes invasion of BRCA2-RAD51 complex into damaged DNA strands by localizing BRCA2 to DSBs [54]. 

Other genes are also suspected of association with FPC onset. In an Italian PC cohort including 16 FPC cases, Ghiorzo et al. found *CDKN2A* germline mutation in 31.3% of FPC patients without *BRCA2* and *PALB2* mutation [55]. Bartsch et al. detected 1100delC variant of *CHEK2* in 2.9% of German FPC families, and Lener et al. also found that relatives harboring this mutation had a 2.3-fold higher PC risk in a Polish cohort [56,57]. Furthermore, van der Heijden et al. or Couch et al. linked mutations in Fanconi anemia genes (*FANCC* and *FANCG*) with young onset PC [58,59]. Pogue-Geile et al. presented palladin (*PALLD*) encoding cytoskeletal component as a novel susceptibility gene, though this was not validated in subsequent studies [60,61]. Finally, Roberts et al. listed spindle-assembly checkpoint gene *BUB1B*, *CPA1* encoding carboxypeptidase A1, *FANCC* and *FANCG* as candidate susceptibility genes based on whole-genome sequencing data for 638 FPC patients [62]. Extensive further studies are required to reveal the functional impact of these genes in FPC.

## 4. Germline Mutation Prevalence in FPC Compared with Sporadic PC

Information about germline mutation in FPC is essential for developing surveillance and treatment strategies, and there have been several studies along these lines (Table 2). According to a germline mutation analysis of four genes (*BRCA1*, *BRCA2*, *PALB2* and *CDKN2A*) in 727 affected cases including 521 FPC, the percentage of cases harboring the above gene mutations was higher in FPC probands than in non-FPC probands (8.0% vs. 3.5%) (odds ratio: 2.4, 95% CI: 1.1–5.4) [63]. In this study, the mutation prevalence in the FPC cohort was 3.7% for *BRCA2*, 2.5% for *CDKN2A*, 1.2% for *BRCA1*, and 0.6% for *PALB2*. Grant et al. conducted targeted sequencing of 13 genes associated with hereditary cancer syndromes or FPC for 290 PC probands, and showed the mutation prevalence was 2.6% (95% CI: 0–6.0) in cases with a history of PC in FDR and 4.0% (95% CI: 2.1–5.9) in those without any history [64]. Petersen et al. also conducted germline analysis of 25 cancer susceptibility genes for 303 PC patients including an FPC cohort in the Mayo Clinic Familial Pancreatic Cancer Registry, and reported that 12.9% of 186 FPC cases and 9.4% of 117 non-FPC cases harbored some mutations [65]. From a Japanese cohort of 1,197 PC patients, Takai et al. identified 88 (7.3%) FPC cases, and conducted germline sequencing of hereditary cancer susceptibility genes [66]. In this study, deleterious mutations were detected in 8 of 54 (14.5%) cases, including 3 (5.6%) *BRCA2* mutation, 2 (3.7%) *PALB2* mutation, 2 (3.7%) *ATM* mutation and 1 (1.9%) *MLH1* mutation. These data indicate genomic heterogeneity in FPC, although inconsistencies in mutation prevalence among studies might be influenced by the number of target genes, limited sample size or subjects’ backgrounds, including ethnicity. Importantly, previous studies found no susceptibility gene mutations in more than 80% of FPC cases, suggesting that further investigations to identify novel susceptibility genes should be fruitful. 

Germline mutations in cancer susceptibility genes are also found in cases without definite family history or association with hereditary syndromes. Hu et al. conducted germline analysis of 22 cancer susceptibility genes for 96 PC cases without preselection based on family history, and detected deleterious mutations in 13.5% of the cases, including 9.4% for four genes (*BRCA1*, *BRCA2*, *ATM*, and *MSH6*) [67]. Yurgelun et al. conducted targeted sequencing of 24 hereditary cancer susceptibility genes in a series of 289 resected PC, and detected mutations in 9.7% of the cases including 7.3% for HRR genes and 1.0% for MMR genes [68]. Targeted sequencing of 32 genes in an unselected PC series of 854 cases revealed deleterious mutations in 3.9% of the cases, including 3.5% for 7 FPC-related genes [69]. Hu et al. recently compared genomic data of 3,030 affected cases and a normal cohort, and found that 5.5% of the affected cases harbored mutation in 6 genes associated with increased PC risk (*CDKN2A*, *TP53*, *MLH1*, *BRCA1*, *BRCA2*, and *ATM*) [70]. In the latest germline analysis of 298 unselected PC patients, 23 (7.7%) patients harbored deleterious mutations in PC susceptibility genes [71]. In this study, 6 of 23 (26.1%) mutated cases actually did not fulfill prescribed genetic testing criteria for hereditary cancer syndrome or FPC, and 12 of the 23 (52.2%) mutated cases would not have been checked according to the criteria. These results indicate that the disparity in germline mutation prevalence between FPC and sporadic PC is not dramatic, and that current screening strategies chiefly based on family history cannot completely extract susceptibility gene carriers. On the other hand, exhaustive genomic analysis seems not to be cost-effective, since mutation is not so frequent. Here, software risk assessment tools based on family and past history of PC, such as PancPRO, might deserve further consideration [72].

## 5. Surveillance Strategy for High-Risk Cases

Considering the low incidence of PC, comprehensive surveillance for the general population is not recommended, and only cases with increased PC risk > 10-fold or lifetime PC risk > 5% are considered as feasible candidates for surveillance [4,73,74]. The expert consensus practice recommendations formulated in the International Symposium of Inherited Diseases of the Pancreas in 2007 describe potential subjects for surveillance as follows: (1) individuals with PJS or hereditary pancreatitis; (2) *BRCA1*, *BRCA2* or *CDKN2A* mutation carriers with at least one affected first- or second-degree relative; (3) individuals with three or more affected first-degree, second-degree or third-degree relatives; and (4) individuals with two affected relatives including at least one FDR [73]. According to the recommendations of the International Cancer of the Pancreas Screening (CAPS) Consortium summit in 2013, surveillance for the following categories is recommended: (1) individuals with PJS; (2) *CDKN2A*, *BRCA2* or MMR gene mutation carriers with at least one affected FDR; and (3) individuals with at least two affected FDRs [74]. Remarkably, descriptions regarding family history are different between the two expert recommendations, and the later CAPS statements list MMR genes instead of *BRCA1*. There are no established protocols for screening modalities, follow-up duration, or time of screening initiation/termination, though Canto proposed a surveillance program by endoscopic ultrasound (EUS) and MRI at 1–3 year intervals from age 40 or from 10 years before the earliest age of PC onset in the family [75].

Vasen et al. selected 134 relatives with two affected FDRs, 80 relatives with at least three affected FDRs, 178 unaffected *CDKN2A* mutation carriers and 19 unaffected *BRCA2*/*PALB2* mutation carriers from three cohorts in Europe, and prospectively monitored them as high-risk individuals using MRI (i.e., magnetic resonance cholangiopancreatography) or EUS [76]. Among *CDKN2A* mutation carriers, 13 (7.3%) cases developed PC, and the clinical outcomes were favorable (75% resection rate and 24% 5-year overall survival (OS) rate). On the other hand, PC occurrence in the FPC cohort was low (0.5%), and the authors concluded that the practical justification for conducting surveillance was not so robust for FPC as for *CDKN2A* mutation carriers. Canto et al. also followed-up 354 unaffected individuals categorized as a high-risk group based on three factors (designated gene mutation, PC family history, or age of PC onset) [77]. In this study, 10 subjects developed PC inside the surveillance program and 4 individuals outside the program, and the OS was significantly better in the former group (3-year OS rate: 85% vs. 25%). According to a systematic review including five prospective controlled studies for familial high-risk individuals, subjects in a screening program, mainly by EUS, had a significantly higher curative resection rate (60% vs. 25%) and longer OS (median OS: 14.5 months vs. 4.0 months) compared with the control group, although psychological function and economic burden were adverse in the screening group [78]. Although the above clinical data are promising, pancreatic resection is a relatively invasive surgery, and the clinical benefit of PC surveillance for high-risk unaffected relatives is unclear owing to no randomized trials. Such programs should be provided as a clinical trial in experienced institutions [75].

## 6. Medication for Patients Harboring Susceptibility Gene Mutations

### 6.1. Platinum-Based Regimen

Several clinical trials have shown clinical benefits of a platinum-based initial regimen, especially for *BRCA*-mutated cases. Golan et al. reviewed clinical outcomes in a series of 71 PC patients harboring *BRCA1*/*BRCA2* mutation, and reported that stage 3/4 patients receiving a platinum regimen had significantly longer OS than those receiving a non-platinum regimen (median OS: 22 months vs. 9 months) [79]. In this study, the combination of gemcitabine (GEM) plus cisplatin (CDDP) was mainly adopted as the platinum regimen. According to a retrospective review of 36 metastatic PC patients receiving the leucovorin calcium, fluorouracil, irinotecan hydrochloride and oxaliplatin (FOLFIRINOX) regimen, cases harboring mutations in DNA damage repair (DDR) genes (*BRCA1*, *BRCA2*, *PALB2*, *MSH2* and *FANCF*) had marginally longer OS than those without the mutations (median OS: 14 months vs. 5 months), and multivariate analysis showed a significant association between DDR gene mutation status and longer OS [80]. The latest NCCN Guidelines for PC describe GEM/CDDP as one of the first-line options for known *BRCA1*/*BRCA2*-mutated cases [81]. Recently, Takahashi et al. presented phase II trial data of the GEM plus oxaliplatin (GEMOX) regimen as a first-line treatment for PC cases [82]. This trial enrolling patients based on personal or family history of pancreatic, breast, ovarian and prostate cancers failed to demonstrate a promising survival benefit (1-year OS rate: 27.9%), which suggests that personal/family history alone might be insufficient for selecting suitable candidates. Randomized trials are needed to establish the position of platinum-based regimens.

### 6.2. PARP Inhibitors

PARP inhibitors have been clinically investigated for *BRCA*-mutated cases as second-line or later treatments (Table 3). This class of agents mechanically inhibits participation of PARP in base excision repair, and causes DSBs due to unrepaired single-strand breaks [83]. DSBs are usually restored by the HRR pathway, whereas BRCA protein deficiency deriving from inactivating *BRCA1*/*BRCA2* mutation inhibits this pathway and finally induces cell death [84,85].

Kaufman et al. conducted a phase II trial of olaparib for 298 patients with advanced *BRCA1*/*BRCA2*-mutated tumors including 23 PC cases [86]. Overall response rate (ORR) was 26.2% in the entire cohort and 21.7% in the PC cohort, and median OS and progression-free survival (PFS) in the PC cohort were 9.8 months and 4.6 months, respectively. The outcomes seem promising for relapsed PC, and this trial has become the foundation for further clinical trials. Shroff et al. performed a phase II trial of rucaparib (RUCAPANC trial) for 19 patients with locally advanced/metastatic PC harboring *BRCA1*/*BRCA2* mutation, and reported an ORR of 21.1% and disease control rate of 31.6% [87]. According to another phase II trial of veliparib for 16 PC patients with *BRCA1*/*BRCA2* mutation, no patient achieved a response, and the median OS and PFS were only 3.1 months and 1.7 months, respectively [88]. The modest efficacy of veliparib as a single agent compared with olaparib or rucaparib is explained by its lower PARP-trapping activity, and combination therapies with cytotoxic agents are expected to become the mainstream of its clinical development [54]. de Bono et al. conducted a phase I trial of talazoparib, which exhibits the strongest PARP-trapping activity for patients with advanced malignancies, and reported that the ORR at 1.0 mg/kg (recommended phase II dose) was 22.2% in the entire cohort and 20.0% in 10 PC patients [89]. Out of two PC cases that achieved a partial response in this trial, one case harbored *BRCA2* mutation and the other, *PALB2* mutation. 

Currently, several clinical trials of PARP inhibitors are underway for advanced PC. Firstly, a phase II trial of rucaparib for locally advanced/metastatic PC cases harboring *BRCA1*, *BRCA2* or *PALB2* mutation is in progress in the Abramson Cancer Center of the University of Pennsylvania, where the clinical efficacy of rucaparib maintenance therapy is being assessed following a platinum-based induction regimen of at least 16 weeks (NCT03140670). In a phase III trial of olaparib (POLO trial), metastatic PC patients with germline *BRCA1*/*BRCA2* mutation who do not experience disease progression after a platinum-based regimen for 16 weeks or more are randomized to olaparib 300 mg twice daily or placebo (NCT02184195). To evaluate combination therapy with a PARP inhibitor, O’Reilly et al. conducted a phase I trial of GEM/CDDP plus veliparib for 17 patients with untreated advanced PC, including 9 cases harboring germline *BRCA1*/*BRCA2* mutation, and reported an ORR of 77.8% and median OS of 23.3 months in *BRCA*-mutated cases [90]. Based on this promising outcome, a randomized phase II trial of GEM/CDDP with or without veliparib is being performed for locally advanced/metastatic PC cases with *BRCA1*, *BRCA2* or *PALB2* mutation (NCT01585805). Currently available evidence for this class of agents is for second-line or later treatment, and the above two randomized trials are expected to provide informative data for initial therapy [91].

Some BRCA-proficient tumors have defects in HRR genes including *ATM*, *ATR*, *CHK1, CHK2, PALB2* and *RAD51*, and these cases, which share the molecular features of *BRCA*-mutant tumors (BRCAness) are considered as good targets for PARP inhibitor treatment [92,93]. Currently, phase II trials of olaparib for BRCAness phenotypic PC are in progress in the M.D. Anderson Cancer Center (NCT02677038) and Sheba Medical Center (NCT02511223).

### 6.3. Immune Checkpoint Inhibitors

While immune checkpoint inhibitors have provided striking clinical benefits for several kinds of malignancies, such as melanoma or non-small-cell lung cancer (NSCLC), the impact of this class of agents on PC as a whole is not evident. The presence of fewer neoantigens than melanoma and NSCLC, and a microenvironment composed of decreased intra-tumoral effector T-cells, increased immunosuppressive immune cells, and abundant extracellular matrix mechanically complicate clinical applications for PC [94,95]. Actually, a phase I trial of anti-PD-L1 antibody identified no responders in a 14 PC cohort in contrast to an ORR of 17% in melanoma or 10% in NSCLC [96].

Previous studies have demonstrated that patients with MMR-deficient colorectal cancer are favorable candidates for immune checkpoint blockade, and screening of MMR-deficient tumors is performed for various kinds of malignancies over colorectal cancer [97,98]. Le et al. evaluated the clinical efficacy of anti-PD-L1 antibody, pembrolizumab, for 86 relapsed patients with MMR-deficient tumors from 12 types of malignancies, including 8 PC patients. They reported that the ORR was 53.4% in the entire cohort and 62.5% in the PC cohort, and the complete response (CR) rate was 20.9% in the former and 25.0% in the latter [99]. The higher response in the PC cohort might be associated with a high mutation burden leading to increased neoantigens or active lymphocyte infiltrates in MMR-deficient tumors [97]. In 2017, the U.S. Food and Drug Administration (FDA) granted accelerated approval to pembrolizumab for patients with advanced MSI-high or MMR-deficient solid tumors, including PC [100].

Here, it should be noted that MMR-deficiency is caused by somatic mutations as well as germline mutations in MMR genes. Humphris et al. immunohistochemically identified 4 MMR-deficient cases from a cohort of 385 sporadic PC patients, and showed that all 4 patients harbored somatic mutations in *MLH1* or *MSH2* without germline mutation [101]. This result highlights the need for screening strategies to find MMR-deficient cases among seemingly sporadic tumors.

## 7. Conclusions and Future Prospects

The overall genomic features and clinical management of hereditary PC syndromes and FPC are summarized in Figure 1. Patients with hereditary cancer syndromes exhibit unique clinical features from a younger age dependent on the particular syndrome, and harbor a specific causative gene. On the other hand, clinical features in FPC cases are ordinary and their genomic backgrounds are heterogenous. The rapid progress in genomic analysis has led to the discovery of several FPC susceptibility genes, such as *BRCA2*, *PALB2* and *ATM*, and investigations to identify further genes are underway.

Exhaustive genomic analyses have shown that surveillance based mainly on family history cannot pick up all susceptibility gene carriers, and more efficient approaches including software risk assessment tools or the use of artificial intelligence are needed. Regardless of some favorable data, the clinical benefits of surveillance programs have not been demonstrated in large-scale trials, and further trials are awaited. In terms of medication, PARP inhibitors for *BRCA*-mutated cases are in the frontline of clinical developments for PC. Randomized trials of olaparib alone or veliparib in combination with cytotoxic agents are underway, and these results are expected to establish the clinical position of PARP inhibitors. In addition, a clinical trial of anti-PD-1 antibody found a marked response for patients with MMR-deficient PC, and further investigation in a larger cohort seems worthwhile.

For the future, there are various avenues to explore. First of all, FPC is an entity especially based on family history, and the genetic backgrounds of these patients are heterogenous, unlike hereditary PC syndromes. Therefore, further discussion is required to treat it as a type of hereditary PC. Secondly, most of the mutations detected in genomic analysis are still undruggable, so development of novel agents is indispensable to link available genomic information with improved outcomes [102,103]. Thirdly, early diagnosis is one of the major goals of surveillance for high-risk cases, though specific approaches for each detected finding remain to be established. Fourthly, the arrival of novel agents presents the practical problem of how to properly use these agents together with conventional cytotoxic agents such as platinum drugs, depending on disease status. Finally, clinical and genomic data about hereditary PC are still limited, especially for non-western populations, and the worldwide spread of the FPC registration system founded in 1994 in the US is expected to provide a solid foundation for future investigations [104].

## Figures and Tables

**Figure 1 ijms-20-00561-f001:**
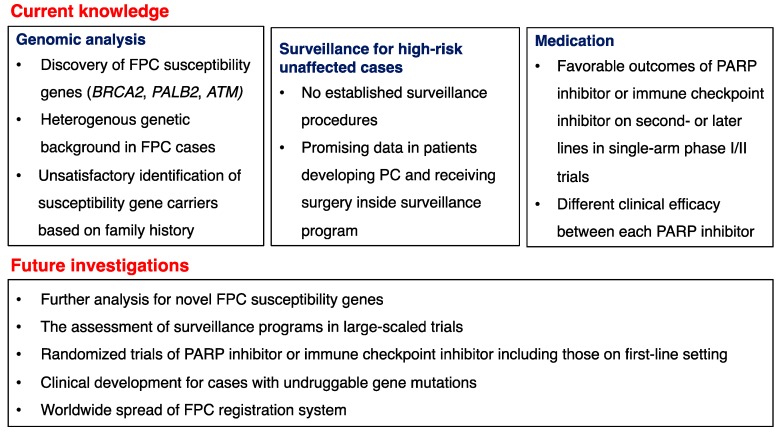
Current status and future prospects for hereditary PC syndromes and FPC from the viewpoints of genomic analysis, surveillance and medication.

**Table 1 ijms-20-00561-t001:** Clinical and germline genomic features in hereditary syndromes.

Syndrome	Clinical Features	Causative Gene	Relative PC Risk	Cumulative PC Risk
PJS	● Perioral/buccal pigmentation ● Gastrointestinal hamartomatous polyp	*STK11/LKB1*	132-fold [9]	11% (70 years) [10] 8% (60 years) [11]
HP	Recurrent acute pancreatitis	*PRSS1*, *SPINK1*	53-fold [14] 87-fold [15]	40% (70 years) [14] 53.5% (75 years) [15] 7.2% (70 years) [16]
FAMMM	Multiple atypical nevi	*CDKN2A*	13.1, 22-fold [22] 46.6-fold [23]	17% (75 years) [21]
FAP	Hundreds of synchronous colorectal adenomas	*APC*, *MUTYH*	4.5-fold [26]	1.7% (80 years) [26]
LS	Nonpolyposis colorectal cancer at early ages	*MLH1*, *MSH2*, *MSH6*, *PMS2*	8.6-fold [31]	3.7% (70 years) [31]
HBOC	Breast and ovarian cancers occurring at early ages	*BRCA1*, *BRCA2*	*BRCA1*:2.6-fold [39] * *BRCA2*:2.1-fold [39] *, 21.7-fold [40] *	NA

PC, pancreatic cancer; PJS, Peutz-Jeghers syndrome; HP, hereditary pancreatitis; FAMMM, familial atypical multiple mole melanoma; FAP, familial adenomatous polyposis; LS, Lynch syndrome; HBOC, hereditary breast and ovarian cancer syndrome; NA, not available. * Past history of breast or ovarian cancer is unavailable.

**Table 2 ijms-20-00561-t002:** Germline mutation analyses for patients with FPC.

Author	Number of Analyzed Patients	Number of Targeted-Genes	Mutation Prevalence	Detected Gene Mutations
Zhen, et al. [62]	521	4	8.0%	*BRCA2* (3.7%), *CDKN2A* (2.5%), *BRCA1* (1.2%), *PALB2* (0.6%)
Grant, et al. [63]	39	13	2.6%	*ATM* (2.6%)
Petersen, et al. [64]	186	25	12.9%	NA
Takai, et al. [65]	54	21	14.5%	*BRCA2* (5.6%), *PALB2* (3.7%), *ATM* (3.7%), *MLH1* (1.9%)

NA, not available.

**Table 3 ijms-20-00561-t003:** Clinical trials of PARP inhibitor for PC patients harboring *BRCA1*/*BRCA2* mutation.

Author	Agent	Phase	Number of Patients	Disease Status	ORR	OS	PFS
Kaufman et al. [84]	Olaparib	II	23	Advanced	21.7%	9.8 months	4.6 months
Shroff et al. [85]	Rucaparib	II	19	Locally advanced, Metastatic	21.1%	NA	NA
Lowery et al. [86]	Veliparib	II	16	Locally advanced, Metastatic	0%	3.1 months	1.7 months
de Bono et al. [87]	Talazoparib	I	10	Advanced	20.0%	NA	NA
O’Reilly et al. [88]	Veliparib plus GEM/CDDP	I	9	Locally advanced, Metastatic	77.8%	23.3 months	NA

ORR, overall response rate; OS, overall survival; PFS, progression-free survival; GEM, gemcitabine; CDDP, cisplatin; NA, not available.

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
