# Peer review of "Genomic Features and Clinical Management of Patients with Hereditary Pancreatic Cancer Syndromes and Familial Pancreatic Cancer"

_ijms, 2019, doi:10.3390/ijms20030561_

Round 1
Reviewer 1 Report
Ohmoto and colleagues reviewed the current our understanding of hereditary pancreatic cancer in association with genetic and clinical features. The paper is very well-written. The reviewer requests some minor revision for improving quality of the manuscript to readers.
1. The authors described mutations of specific genes in the hereditary pancreatic cancer syndromes. However, the authors did not specify whether those mutations are germline or sporadic. For better understanding of the readers, the authors should specify those mutations throughout their manuscript.
2. Although this paper is review article, the authors described unnecessary p-values in many part of their manuscript. The reviewer recommend to delete unnecessary p-values (for example, lines 74, 291, 293, 294, and 308) and re-describe the sentence without citing p-value from the references.
3. In line 123, the authors described that “Patients with LS suffer from colorectal cancer at early age.” Comparing with sporadic cancers how younger were patients? The authors should describe more specifically.
4. On the same line, the authors described that “HBOC is a syndrome characterized by breast and ovarian cancers occurring at early ages” in line 151. Comparing with sporadic cancers how younger were patients? The authors should describe more specifically.
5. The descriptions in lines 295-298 are not well matched with the flow with other sentences in the section. The reviewer recommends reorganize the description.
Author Response
Comment 1. The authors described mutations of specific genes in the hereditary pancreatic cancer syndromes. However, the authors did not specify whether those mutations are germline or sporadic. For better understanding of the readers, the authors should specify those mutations throughout their manuscript.
Response: Thank you so much for the important comment. Genomic mutations described at the part of hereditary pancreatic cancer syndromes are germline mutations. Therefore, we have added the following sentence on page 2, lines 59 as follows:
“Representative hereditary syndromes with specific germline mutations are summarized in Table 1.”
Comment 2. Although this paper is review article, the authors described unnecessary p-values in many parts of their manuscript. The reviewer recommend to delete unnecessary p-values (for example, lines 74, 291, 293, 294, and 308) and re-describe the sentence without citing p-value from the references
Response: Thank you so much for the comment. We have deleted p-values on page 7, lines 291, 293, 294, 305, 310, 311.
Comment 3. In line 123, the authors described that “Patients with LS suffer from colorectal cancer at early age.” Comparing with sporadic cancers how younger were patients? The authors should describe more specifically.
Response: Thank you so much for the important comment. As recommended, we have added detailed age of the onset in patients with LS on page 3, lines 122-123 as follows:
“at early ages (typically in the mid 40 years about 20 years younger compared with sporadic cases)”.
Comment 4. On the same line, the authors described that “HBOC is a syndrome characterized by breast and ovarian cancers occurring at early ages” in line 151. Comparing with sporadic cancers how younger were patients? The authors should describe more specifically.
Response: Thank you so much for the comment. As recommended, we have described detailed age of the onset in patients with HBOC on page 4, lines 151 as follows:
“before the age of 50”.
Comment 5. The descriptions in lines 295-298 are not well matched with the flow with other sentences in the section. The reviewer recommends reorganize the description.
Response: Thank you so much for the great comment. As the reviewer points out, general explanation about surgical indication seems unsuitable for the flow. Therefore, we have deleted the sentence and changed the next sentence on page 7, lines 295-297 as follows:
“Although the above clinical data are promising, pancreatic resection is a relatively invasive surgery, and the clinical benefit of PC surveillance for high-risk unaffected relatives is unclear owing to no randomized trials.”
Reviewer 2 Report
The paper is an overview of published knowledge. As such it could serve as an updated overview of literature which may be of interest for researchers in the various fields of inherited cancer syndromes.
The shortcoming is the claim that there is a what is called familial pancreatic cancer which comes in addition to the inherited cancer syndromes in which pancreatic cancer is part. This is so because none of these syndromes are diagnosable by family history (positive and negative predictive values to do so are very low) and as discussed there are no evidence from genetic testing that there are more genes causing inherited pancreatic cancer, and there is no epidemiological evidence that there is an excess of familial clustering besides the known inherited cancer syndromes, which again is because we do not know the prevalence of these syndromes and a 'negative family history' has no substantial negative predictive value to exclude such. Thus, the claim that there is a site-specific inherited pancreatic cancer phenotype in addition to the the cancer syndrome known is not substantiated. If the mns is revised to be an overview of current knowledge of pancreatic cancer associated with known inherited cancer syndromes, I leave to the editor to decide whether or not such an overview is within their priority to publish.
Author Response
Comment 1. The shortcoming is the claim that there is a what is called familial pancreatic cancer which comes in addition to the inherited cancer syndromes in which pancreatic cancer is part. This is so because none of these syndromes are diagnosable by family history (positive and negative predictive values to do so are very low) and as discussed there are no evidence from genetic testing that there are more genes causing inherited pancreatic cancer, and there is no epidemiological evidence that there is an excess of familial clustering besides the known inherited cancer syndromes, which again is because we do not know the prevalence of these syndromes and a 'negative family history' has no substantial negative predictive value to exclude such. Thus, the claim that there is a site-specific inherited pancreatic cancer phenotype in addition to the cancer syndrome known is not substantiated. If the mns is revised to be an overview of current knowledge of pancreatic cancer associated with known inherited cancer syndromes, I leave to the editor to decide whether or not such an overview is within their priority to publish.
Response: Thank you so much for the essential comment. As the reviewer comments, familial pancreatic cancer (FPC) is defined only based on PC family history, and genetic backgrounds in FPC patients are heterogenous. In that sense, it is still controversial whether we could treat it as a type of hereditary PC, and we agree with the idea that FPC has not been a genetically established entity. On the other hand, since MacDermott et al. reported four PC cases out of six siblings, an extraordinary PC occurrence in some pedigrees has been revealed (Gastroenterology 1973; 65: 137―139). As Klein et. showed (References 43), PC family history in first-degree relatives (FDRs) is considered as one of the independent PC risk factors, where the risk becomes higher in cases with strong family history. In terms of miserable prognosis of PC, highlighting these unaffected relatives with the family history as a high-risk group seems reasonable. Further genomic investigations for FPC are enthusiastically underway, and as we described at the section of surveillance strategy for high-risk cases on page 6-7 (References 73, 74, 76, 77), surveillance programs generally target cases with strong family history as well as those with hereditary PC syndromes. Therefore, the latest review about FPC as well as hereditary syndromes seems informative.
To avoid confusion, we have separated FPC from hereditary PC syndromes obviously, and changed a part of the title from “hereditary pancreatic cancer” into “hereditary pancreatic cancer syndromes and familial pancreatic cancer”. Similarly, we have changed the word “hereditary PC” on page 2, lines 49-50 and page 9, lines 389. Moreover, we have added two sentences about current status of FPC at the part of future prospects on page 10, lines 406- 408 as follows:
“First of all, FPC is an entity especially based on family history, and genetic backgrounds in these patients are heterogenous unlike hereditary PC syndromes. Therefore, further discussion is required to treat it as a type of hereditary PC.”
Round 2
Reviewer 2 Report
The responses to my initial comments are satisfactory.